# Construction of Soybean Mutant Diversity Pool (MDP) Lines and an Analysis of Their Genetic Relationships and Associations Using TRAP Markers

**Dong-Gun Kim [1,2,†], Jae Il Lyu [1,†], Min-Kyu Lee [1,3], Jung Min Kim [1,3], Nguyen Ngoc Hung [1,3], Min Jeong Hong [1], Jin-Baek Kim [1], Chang-Hyu Bae [2,*] and Soon-Jae Kwon [1,*]**

[1] Advanced Radiation Technology Institute, Korea Atomic Energy Research Institute, Jeongup 56212, Korea; dgkim@kaeri.re.kr (D.-G.K.); jaeil@kaeri.re.kr (J.I.L.); biolmk@kaeri.re.kr (M.-K.L.); jmkim0803@kaeri.re.kr (J.M.K.); nguyenhung@kaeri.re.kr (N.N.H.); hongmj@kaeri.re.kr (M.J.H.); jbkim74@kaeri.re.kr (J.-B.K.)

[2] Department of Life-resources, Graduate School, Sunchon National University, Suncheon 57922, Korea

[3] Division of Plant Biotechnology, College of Agriculture and Life Science, Chonnam National University, Gwangju 61186, Korea

**\*** Correspondence: soonjaekwon@kaeri.re.kr (S.-J.K.); chbae@scnu.ac.kr (C.-H.B.); Tel.: +82-63-570-3312 (S.-J.K.); +82-61-750-3214 (C.-H.B.)

**†** These authors contributed equally to this work.

**Abstract:** Mutation breeding is useful for improving agronomic characteristics of various crops. In this study, we conducted a genetic diversity and association analysis of soybean mutants to assess elite mutant lines. On the basis of phenotypic traits, we chose 208 soybean mutants as a mutant diversity pool (MDP). We then investigated the genetic diversity and inter-relationships of these MDP lines using target region amplification polymorphism (TRAP) markers. Among the different TRAP primer combinations, polymorphism levels and polymorphism information content (PIC) values averaged 59.71% and 0.15, respectively. Dendrogram and population structure analyses divided the MDP lines into four major groups. According to an analysis of molecular variance (AMOVA), the percentage of inter-population variation among mutants was 11.320 (20.6%), whereas mutant intra-population variation ranged from 0.231 (0.4%) to 14.324 (26.1%). Overall, intra-population genetic similarity was higher than that of inter-populations. In an analysis of the association between TRAP markers and agronomic traits using three different statistical approaches based on the single factor analysis (SFA), the Q general linear model (GLM), and the mixed linear model (Q+K MLM), we detected six significant marker–trait associations involving five phenotypic traits. Our results suggest that the MDP has great potential for soybean genetic resources and that TRAP markers are useful for the selection of soybean mutants for soybean mutation breeding.

**Keywords:** mutation breeding; soybean; mutant diversity pool (MDP); TRAP markers; association analysis

## 1. Introduction

Soybeans (*Glycine max* L.), used for food, livestock feed, and biofuel, is one of the most important agricultural crops worldwide. Soybeans are consumed directly by humans, especially in many Asian countries, in the form of traditional food products such as tofu, soy flour, and soymilk [1,2]. Soybean seeds are composed of 40%–42% protein, 18%–22% oil (85% unsaturated and 15% saturated fatty acids), 28% carbohydrates, and abundant quantities of other nutrients, such as phosphorus, calcium, iron,

lysine, and vitamins A, B, and D [3]. In addition, soybeans play an important role in crop diversification and improve other crops through its addition of nitrogen to the soil during crop rotation [4].

Because the rate of spontaneous mutations in higher plants is quite low ($10^{-5}$ to $10^{-8}$) [5], physical and chemical mutagens can be used to induce mutations in cultivated plants [6]. Gamma radiation is a very effective tool to induce genetic variation in many plant characters, with the resulting changes dependent on the irradiation dose. Various plant organisms, such as seeds, pollen, whole plants, and embryoid bodies, can be irradiated [7]. Because gamma rays can also cause various types of DNA damage, including single- or double-strand breaks and substitutions [8,9], agronomic traits, such as flowering, maturation date, seed coat color, chloroplast number, and biomass yield, are frequently altered in soybean [10,11]. At present, 3200 mutant varieties of more than 210 plant species have been produced for commercial use. Approximately 170 mutant varieties of soybean, the second-most registered species after rice, are found in the FAO/IAEA Mutant Variety Database (http://mvd.iaea.org).

The use of molecular marker-based techniques in genetic studies, such as estimation of genetic diversity and population structure, has advanced remarkably in recent years. Among the different types of DNA markers, restriction fragment length polymorphisms (RFLPs), random amplified polymorphic DNAs (RAPDs), amplified fragment length polymorphisms (AFLPs), and inter-simple sequence repeats (ISSRs) have been extensively used in soybeans, each with their own advantages and limitations [12]. In addition, SNPs, which are widely distributed throughout genomes in both non-coding and coding regions, constitute the most abundant molecular markers recently used in plant genetic breeding [13], but their development is time-consuming and costly. The target region amplification polymorphism (TRAP) is a relatively new, simple, polymerase chain reaction (PCR)-based marker system that takes advantage of the available EST database sequence information to generate polymorphic markers targeting candidate gene [14]. Essentially, it derives an 18-mer primer from the EST sequence and pairs it with an arbitrary primer that targets the intron and/or exon region (AT- or GC-rich core). Because it can be used to generate markers for specific gene sequences, the TRAP technique is useful for genotyping germplasm and generating markers associated with desirable crop agronomic traits for marker-assisted breeding [15]. In recent years, the TRAP marker technique has been applied for genetic diversity analyses [16,17] and genetic mapping [18]. In addition, Im et al. [19] have developed a transposable element-based TRAP (TE-TRAP) marker system that is reportedly suitable for the mutation breeding of sorghum. Although TRAP markers have most commonly been used for genetic mapping and phylogenetic studies, they have also recently been applied to detect DNA mutations [20].

Rapid advances in the field of molecular biology and its allied sciences have led to the routine use of molecular markers, thereby providing plant breeders with a precise genetic-diversity analysis tool for plant improvement [21,22]. A combined molecular and morphological analysis is one of the most widely used approaches for the estimation of genetic distances within a group of genotypes, and molecular markers serve as an excellent tool for obtaining genetic information. Molecular markers are also of great value to plant breeders for assessment of genetic divergence among genotypes for various agronomic traits [23]. Another recent strategy for analyzing agronomic traits, association analysis based on molecular-marker linkage disequilibrium (LD), can reduce experimental time and costs. Association analysis has therefore been widely applied to study a variety of crops, such as rice [24], maize [25], and soybeans [26].

In this study, we constructed 208 mutant diversity pool (MDP) lines based on agronomic traits and investigated their genetic diversity and relationships using TRAP markers. Finally, we performed an association analysis between agronomic traits and polymorphic TRAP amplicons.

## 2. Materials and Methods

### 2.1. Plant Materials and Phenotypic Evaluation

A total of 1000 seeds each of one soybean landraces, KAS360-22, and six representative Korean soybean cultivars [27], 94Seori, BangSa (BS), PalDal (P), DanBaek (DB), DaePung (DP), and HwangKeum

(HK), were irradiated with 250 Gy of gamma rays using a $^{60}$Co gamma-irradiator (150 TBq capacity; ACEL, Ottawa, ON, Canada) at the Korea Atomic Energy Research Institute in 2008. The irradiated $M_1$ and control (non-irradiated) seeds were immediately sown in the research field of the Advanced Radiation Technology Institute. To construct MDP lines, we sowed 1000 irradiated seeds of each of the seven soybean cultivars and harvested a total of 1695 $M_{1:2}$ individual seeds, only excluding those exhibiting growth aberrations, such as stunted growth, pollen sterility, and no germination due to the degree of radiosensitivity (Figure 1). Next, we generated 1695 individual gamma-irradiated mutants during $M_1$–$M_5$ generations by single-seed descent and then continued as bulks until $M_{12}$ generation. In a first selection phase, we selected 523 mutant lines from the $M_5$ generation that possessed at least 30% superior agricultural characteristics related to various environmental factors, such as grain yield, growth type, and climate adaptability. In a second selection phase, we investigated the morphological phenotypes of the 523 mutant lines in the $M_{12}$ generation to eliminate redundant phenotypes. Overall, we selected 208 genetically fixed mutant lines (201 mutants with their wild types), which we designated as the mutant diversity pool (MDP), and assessed the following four agronomic traits: days of flowering (DF), maturation date (MD), seed index (SI), node number (NN), and the following seven morphological traits: growth type (GT), flower color (FC), seed coat color (SCC), seed hilum color (SHC), stem anthocyanin (SA), plant height (PH), and ramification number (RN).

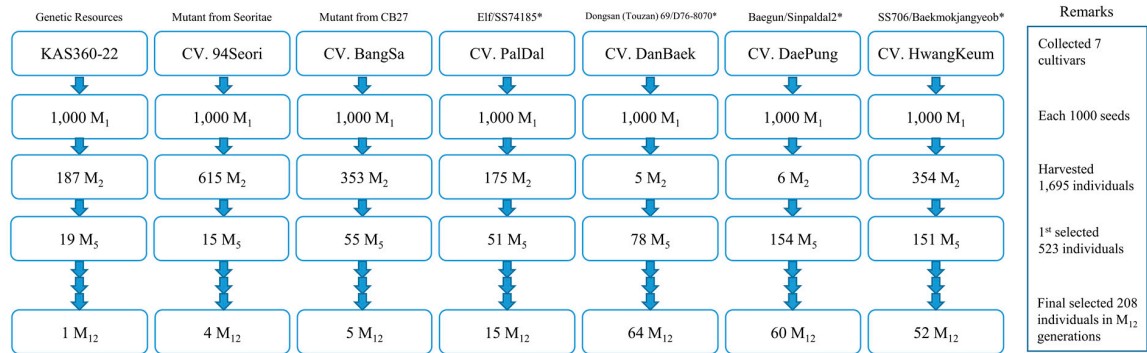

**Figure 1.** Schematic illustration of the breeding of $M_1$–$M_{12}$ generations of 208 soybean mutant diversity pool (MDP) lines. * Information of cultivars was described in Lee et al. [27]

*2.2. DNA Extraction*

The control and treated seeds of the 208 genetically fixed mutant lines were immediately sown in 50-cell (5 × 10) vegetable nursery trays containing bio-bed soil (Dongbu Farm Hannong, Gimje, Korea) and then incubated in a greenhouse at 20 ± 5 °C under natural light for 1 month. Fresh leaf tissue from seedlings of each mutant line was collected and subjected to total genomic DNA extraction using a DNeasy 96 Plant kit (Qiagen, Leipzig, Germany) following the manufacturer's protocol. The extracted DNAs were stored at −20 °C until use. For PCR analysis, DNA concentrations were determined using a NanoDrop ND-1000 spectrophotometer (Thermo Fisher Scientific., Waltham, MA, USA) were and then adjusted to 10 ng/μl.

*2.3. TRAP Analysis*

Four fixed primers and four arbitrary primers were used to generate TRAP markers (Table 1). All four arbitrary primers and one of the fixed primers were designed from other studies of monocot plants [14,16]. The three other fixed primers (with 'MIR' prefixes) were designed based on *Arabidopsis thaliana* microRNA sequences [28] using the Primer3 program (http://frodo.wi.mit.edu/primer3/). PCR amplifications with 16 primer combinations were carried out on all DNA samples according to the protocol of Hu et al. [29] with slight modification. Briefly, reactions were performed in 20-μl volumes containing 2 μl genomic DNA (10 ng/μl), 1 μl fixed primer (10 pmol/μl), 1 μl of each arbitrary primer (10 pmol/μl), 0.8 μl of dNTPs (2.5 mM), 2.0 μl 10 × PCR buffer, and 0.3 μl Phoenix *Taq* DNA polymerase

(5 U/μl; cat no. Phoenix2013). DNA amplification was performed in a thermocycler (G Storm, UK) according to the following program: initial denaturation at 94 °C for 2 min, followed by 5 cycles of 94 °C for 45 s, 35 °C for 45 s, and 72 °C for 60 s, then 35 cycles of 94 °C for 45 s, 53 °C for 45 s, and 72 °C for 60 s, and a final extension at 72 °C for 7 min. The amplified products were analyzed separately using a fragment analyzer automated capillary electrophoresis instrument (FA; Advanced Analytical Technologies, Ankeny, USA), and the collected images were scored manually.

**Table 1.** List of soybean target region amplification polymorphism- (TRAP) marker primers.

| Primer name | Sequence (5′–3′) |
| --- | --- |
| Fixed primers | |
| B14G14B | AAT CTC AAG GAC AAA AGG |
| MIR 156A | GAT CTC TTT GGC CTG TC |
| MIR 157B | GAT CAT TGT CCA GAT TC |
| MIR 159A | GAT CCT TGG TTC TTT GG |
| Arbitrary primers | |
| Sa4 | TTA CCT TGG TCA TAC AAC ATT |
| Sa12 | TTC TAG GTA ATC CAA CAA CA |
| Ga3 | TCA TCT CAA ACC ATC TAC AC |
| Ga5 | GGA ACC AAA CAC ATG AAG A |

## 2.4. Data Analysis

TRAP marker alleles were scored as binary data, with '0′ indicating the absence of a given allele and '1' indicating its presence. The binary data were entered into a Microsoft Excel spreadsheet, and genetic similarities were computed. Using the 0/1-matrix, we calculated gene diversity, percent polymorphism, polymorphism information content (PIC), and genetic distance with the genetic analysis package PowerMarker. A dendrogram was constructed according to the unweighted pair group method with arithmetic mean (UPGMA) algorithm based on the Nei distance method in PowerMarker v3.25 as well as the embedded MEGA7 program. To analyze population structure, a Bayesian population analysis was performed in STRUCTURE 2.3.4. A graphical determination of the optimal number of populations ($K$) was carried out using STRUCTURE HARVESTER (http://taylor0.biology.ucla.edu/structureHarvester/). The number of putative populations ($K$), assumed according to a set of allele frequencies at each locus, can provide the degree of admixture of mutant lines. Each individual was assigned to one or more populations based on membership ($q$) using a Q-matrix derived from STRUCTURE 2.3.4. Ideally, the average estimated log probability of the data Pr(x|k) should plateau at the most appropriate value of $K$. To determine the optimal $K$, we calculated the average probability of $K$ for values of $K = 2 − 15$ based on 10 Markov chain Monte Carlo runs, each consisting of 10,000 burn-in (initiation) iterations followed by 100,000 iterations under a population admixture model. We then conducted an analysis of molecular variance (AMOVA) and calculated genetic distances to support the genetic diversity information. An AMOVA of 999 permutations was completed to assess inter- and intra-population variance (wild type and mutants) using GenALEx v6.501. Pairwise fixation index (Fst) values were also estimated by AMOVA. The hypothesis of an association of molecular markers with phenotypic data was tested using three different methods in the software program TASSEL 4. The first association method, a single factor analysis (SFA) of variance that does not consider population structure, was performed using each marker as the independent variable. The mean performance of each allelic class was compared using the general linear model (GLM) function in TASSEL. Next, a Q GLM method was carried out using the same software. This method applies population structure detected by STRUCTURE (Q matrix) as co-factors. To obtain an empirical threshold for marker significance and an experiment-wise $P$-value, 10,000 permutations of the data were performed. The final marker–trait association test, Q + K MLM, considers both the kinship matrix and the population structure Q-matrix. The K matrix of pairwise kinship coefficients for all pairs of lines was calculated from the TRAP data by TASSEL. Basic statistics and a correlation analysis of agronomic traits were performed using Microsoft Excel and Python v2.7.

## 3. Results

### 3.1. Phenotypic Analysis and Correlation Analysis

A summary of agronomic and morphological traits of the 208 MDP lines is shown in Table 2 and Supplementary Table S1. With respect to growth characteristics, a variety of phenotypes were observed. DF ranged from 42 (mutant numbers; S87, S88, and S149) to 64 (S138), while MD varied from 112 (S6) to 150 (S8). In addition, the seed index (SI) ranged from 7.9 (S13) to 28.8 (S7) g, and PH ranged from 23 (S14) to 92.2 (S76) cm. NN and RN varied from 8.4 (S14) to 24.6 (S78) and 2.6 (S81) to 8.8 (S12) cm, respectively. As shown in the histogram in Figure 2, the phenotypic values of these six quantitative traits in the 208 soybean MDP lines followed a Gaussian distribution. The SI was relatively well distributed, whereas DF and MD had dynamic distributions. Substantial variation was observed in agronomic traits between mutants and wild types of the 208 MDP lines (Figure 3, Figure S1). Compared with their wild types, P- and DB-derived mutants possessed a wider distribution of PH and NN phenotypes, whereas the phenotypes of the six quantitative traits of the HK-derived mutants were only slightly changed. DB-derived mutants, in particular, had mostly increased values of agronomic traits, such as DF, SI, PH, NN, and RN, compared with the wild type. In contrast, HK-derived mutants tended to have reduced values relative to the wild type, albeit only very slightly smaller (Figure 3). With regards to the five qualitative traits, altered phenotypes, such as changes in growth type and color-related traits, were confirmed in the MDP lines (Figure S1). The seven wild-type plants exhibited determinate growth, whereas 46 P-, DB-, and DP-derived mutants were indeterminate. In addition, changes were observed in color-related traits, including FC, SCC, SHC, and SA. These results indicate that MDP lines were successfully constructed through multiplex genetic and phenotypic mutation induced by gamma irradiation. In addition, we calculated the pairwise correlation coefficients of the 11 agronomic traits in the 208 soybean MDP lines. The strongest positive correlations were between PH and NN (0.912*), GT and NN (0.824*), and GT and PH (0.749*), while the most negative correlations were those between MD and SI (−0.512*) and between SI and NN (−0.357*) (Table 3).

**Table 2.** Summary of quantitative trait values in 208 soybean MDP lines.

| Values | Agronomic Traits | | | | Morphological Traits | |
|---|---|---|---|---|---|---|
| | Days of Flowering | Maturity Days | Seed Index (g) | Node Number (ea) | Plant Height (cm) | Ramification Number (ea) |
| Min | 42 | 112 | 7.9 | 8.4 | 23.0 | 2.6 |
| Mean | 52.83 | 133.58 | 18.06 | 13.40 | 42.73 | 4.96 |
| Max | 64 | 150 | 28.8 | 24.6 | 92.2 | 8.8 |
| Line No. | | | | | | |
| Min | S87, S88, S149 | S6 | S13 | S14 | S14 | S81 |
| Max | S138 | S8 | S7 | S78 | S76 | S12 |

Min, minimum; max, maximum; Line No., see Supplementary Table S1. Data were investigated in 2019 at the KAERI research field, Jeongup, Korea. Traits are all means of five biological replicates.

**Table 3.** Matrix of correlation coefficients between 11 agronomic traits in 208 soybean MDP lines.

| | DF | MD | GT | FC | SCC | SHC | SI | SA | PH | NN | RN |
|---|---|---|---|---|---|---|---|---|---|---|---|
| DF (days of flowering) | – | 0.419 * | 0.266 * | 0.436 * | 0.038 | 0.060 | −0.327 * | 0.435 * | 0.272 * | 0.323 * | 0.248 * |
| MD (Maturity days) | | – | 0.269 * | 0.369 * | 0.042 | 0.034 | −0.512 * | 0.491 * | 0.407 * | 0.381 * | 0.224 * |
| GT (Growth type) | | | – | 0.031 | 0.129 * | 0.237 * | 0.308 * | 0.117 | 0.749 * | 0.824 * | 0.101 |
| FC (Flower color) | | | | – | 0.301 * | 0.280 * | 0.369 * | 0.685 * | 0.058 | 0.003 | 0.099 |
| SCC (Seed coat color) | | | | | – | 0.354 * | 0.071 | 0.243 * | 0.166 * | 0.134 * | 0.068 |
| SHC (Seed hilum color) | | | | | | – | 0.005 | 0.197 * | 0.241 * | 0.260 * | 0.124 * |
| SI (Seed index) | | | | | | | – | 0.554 * | −0.295 * | −0.357 * | −0.211 * |
| SA (Stem anthocyanin) | | | | | | | | – | 0.027 | 0.103 | 0.009 |
| PH (Plant height) | | | | | | | | | – | 0.912 * | 0.177 * |
| NN (Node number) | | | | | | | | | | – | 0.202 * |
| RN (Ramification number) | | | | | | | | | | | – |

* Significant at the 0.05 probability level.

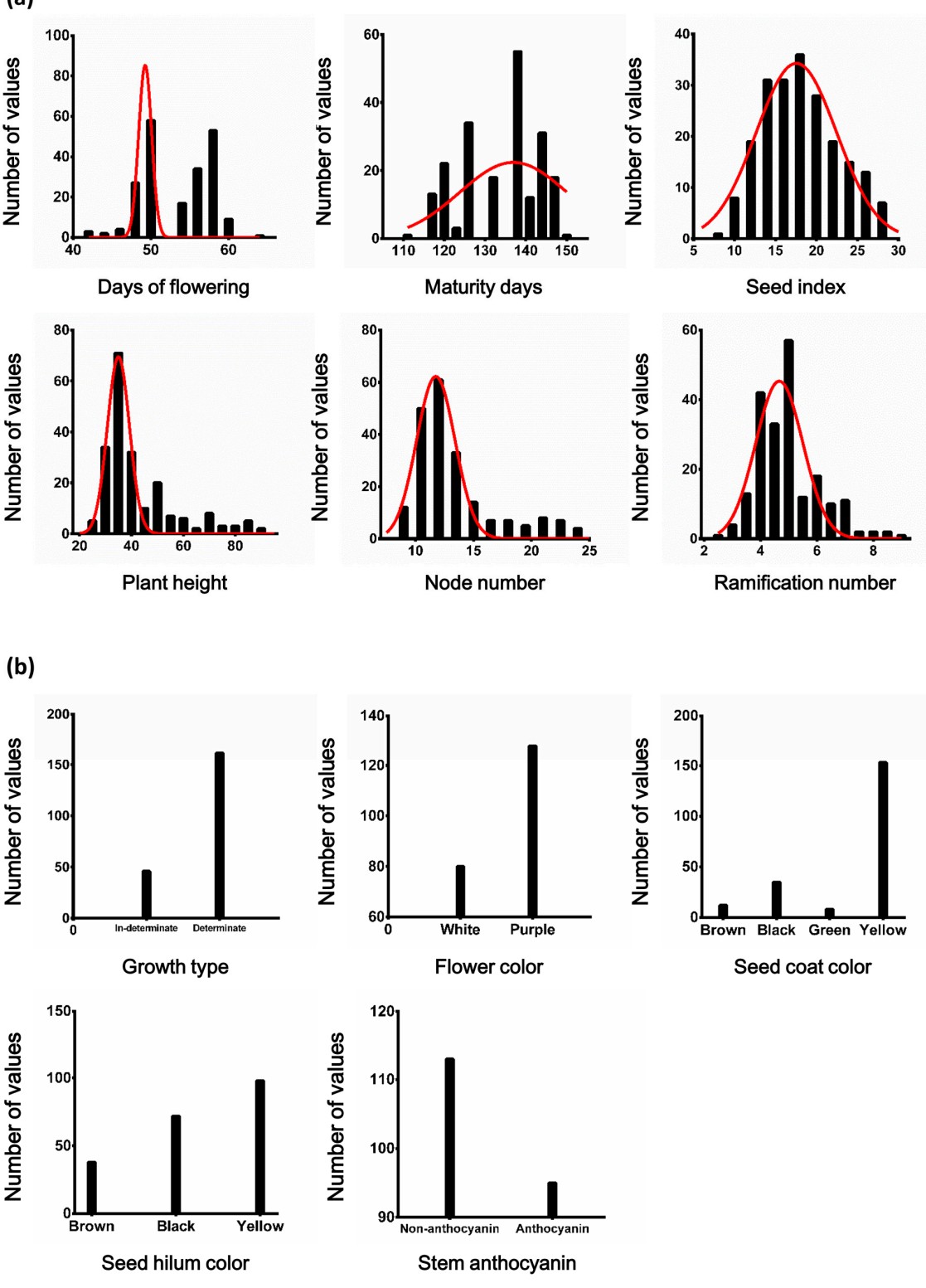

**Figure 2.** Distribution of agronomic traits among 208 soybean MDP lines. Data are presented for (**a**) six quantitative traits (DF, MD, SI, PH, NN, and RN) with gaussian fitting curve and (**b**) five qualitative traits (GT, FC, SCC, SHC, and SA).

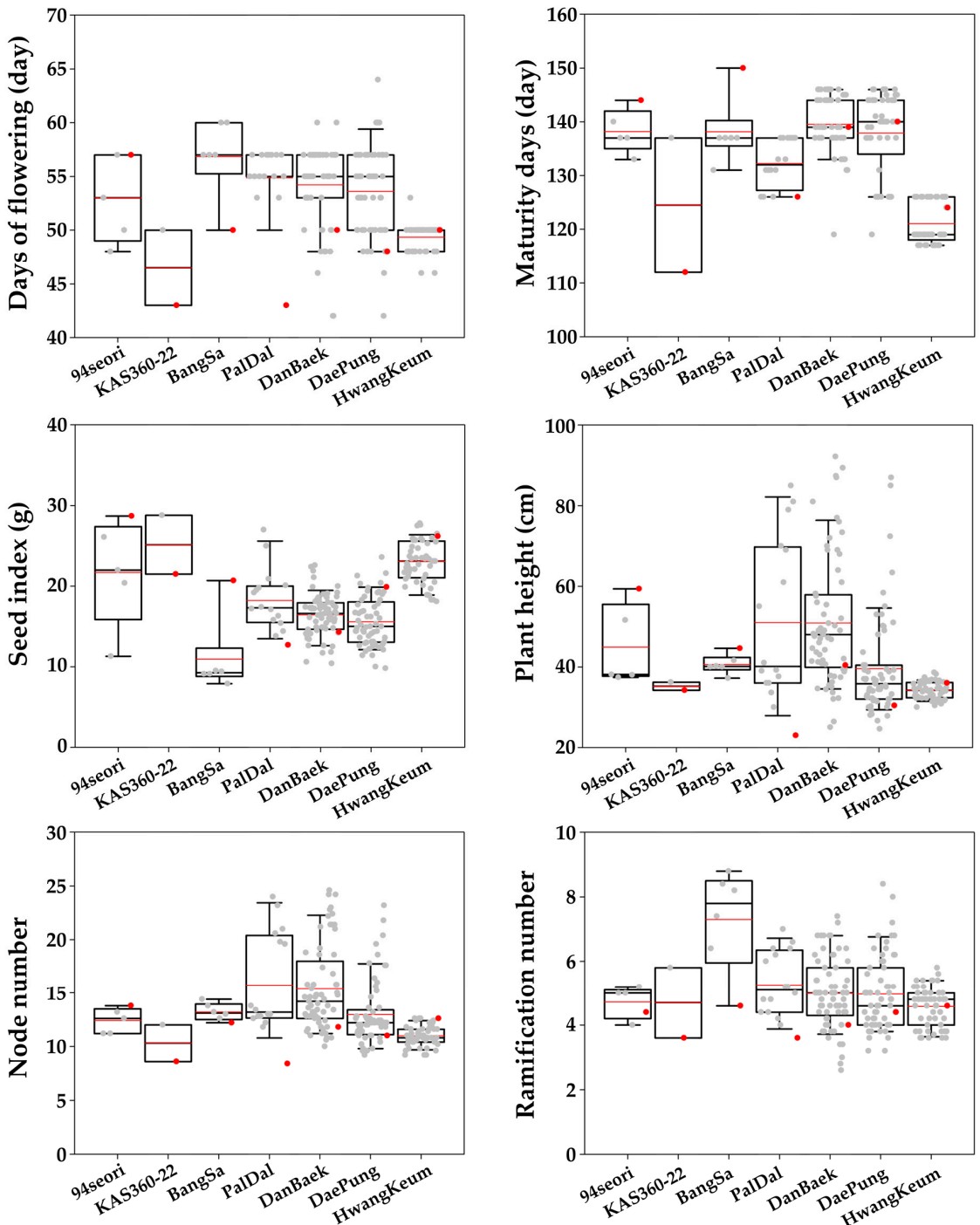

**Figure 3.** Changes in the phenotypes of six quantitative traits in 208 MDP lines. The box plots of phenotypic distributions in seven MDP lines and their wild types are shown. The data shown are the mean values of individual mutants (gray) and wild types (red).

## 3.2. TRAP Marker Polymorphism

A summary of the TRAP markers produced by 16 primer combinations (four fixed forward primers in combination with arbitrary reverse primers) across all 208 soybean mutants is given in Table 4. Sixteen primer combinations amplified a total of 551 fragments. The number of amplified fragments ranged from 25 (for primers MIR157B + Ga5) to 45 (for primers B14G14B + Ga5). A total of 551 amplicons were scored, of which 222 (40.29%) were monomorphic alleles and 329 (59.71%) were

polymorphic. An average of 34.44 amplicons, 20.56 polymorphic, were scored per primer combination. The highest (84.00%) and lowest (32.35%) polymorphism levels were obtained with primers MIR157B + Ga5 and B14G14B + Ga3, respectively. PIC varied among the primer combinations, ranging from 0.07 (B14G14B + Sa12) to 0.23 (MIR157B + Sa4), with a mean value of 0.15.

**Table 4.** Summary of polymorphism of 16 TRAP marker sets in 208 soybean MDP lines.

| Primer Combination | Total Number of Fragments | Polymorphic Fragments | Polymorphism (%) | PIC |
|---|---|---|---|---|
| B14G14B + Sa4 | 38 | 23 | 60.53 | 0.15 |
| B14G14B + Sa12 | 38 | 17 | 44.74 | 0.07 |
| B14G14B + Ga3 | 34 | 11 | 32.35 | 0.08 |
| B14G14B + Ga5 | 45 | 29 | 64.44 | 0.16 |
| MIR156A + Sa4 | 39 | 31 | 79.49 | 0.16 |
| MIR156A + Sa12 | 37 | 22 | 59.46 | 0.16 |
| MIR156A + Ga3 | 35 | 17 | 48.57 | 0.12 |
| MIR156A + Ga5 | 35 | 25 | 71.43 | 0.20 |
| MIR157B + Sa4 | 31 | 23 | 60.53 | 0.23 |
| MIR157B + Sa12 | 31 | 19 | 61.29 | 0.16 |
| MIR157B + Ga3 | 30 | 17 | 56.67 | 0.14 |
| MIR157B + Ga5 | 25 | 21 | 84.00 | 0.21 |
| MIR159A + Sa4 | 36 | 22 | 58.33 | 0.18 |
| MIR159A + Sa12 | 37 | 18 | 48.65 | 0.10 |
| MIR159A + Ga3 | 29 | 15 | 51.72 | 0.16 |
| MIR159A + Ga5 | 31 | 20 | 64.52 | 0.16 |
| Total | 551 | 329 | – | – |
| Average | 34.44 | 20.56 | 59.71 | 0.15 |

### 3.3. Genetic Relationships and Population Structure of the 208 MDP Lines

A dendrogram was constructed to clarify genetic relationships of the 208 MDP lines. At a genetic distance of 0.097, the seven wild-type cultivars and their mutants could be divided into four major groups (Figure 4). Group I included five mutants with their wild types KAS360-22 and 94seori. Group II comprised 22 mutants originating from BS and P. Group III was made up of two subgroups: III-a, which mainly contained DP mutants and their wild type DP as well as some HK mutants, and III-b, which mainly included HK mutants and HK with a few DP mutants. Group IV was distinct from the other three groups and consisted of all 64 DB mutants with DB and DP mutants. We performed a population structure analysis with a predefined number of sub-populations ($K$) ranging from 2 to 15. The optimal $K$ was determined using an ad-hoc statistic ($\Delta K$), which was based on the rate of change in the log probability of the data between successive $K$-values (Figure S2). According to the analysis, the optimal $K$ value was 4, which corresponded to a division of the genetic composition into four groups (Figure 4). Each accession was assigned to single or multiple membership depending on whether its genotype indicated admixture. The result of this analysis was consistent with the topology of the dendrogram.

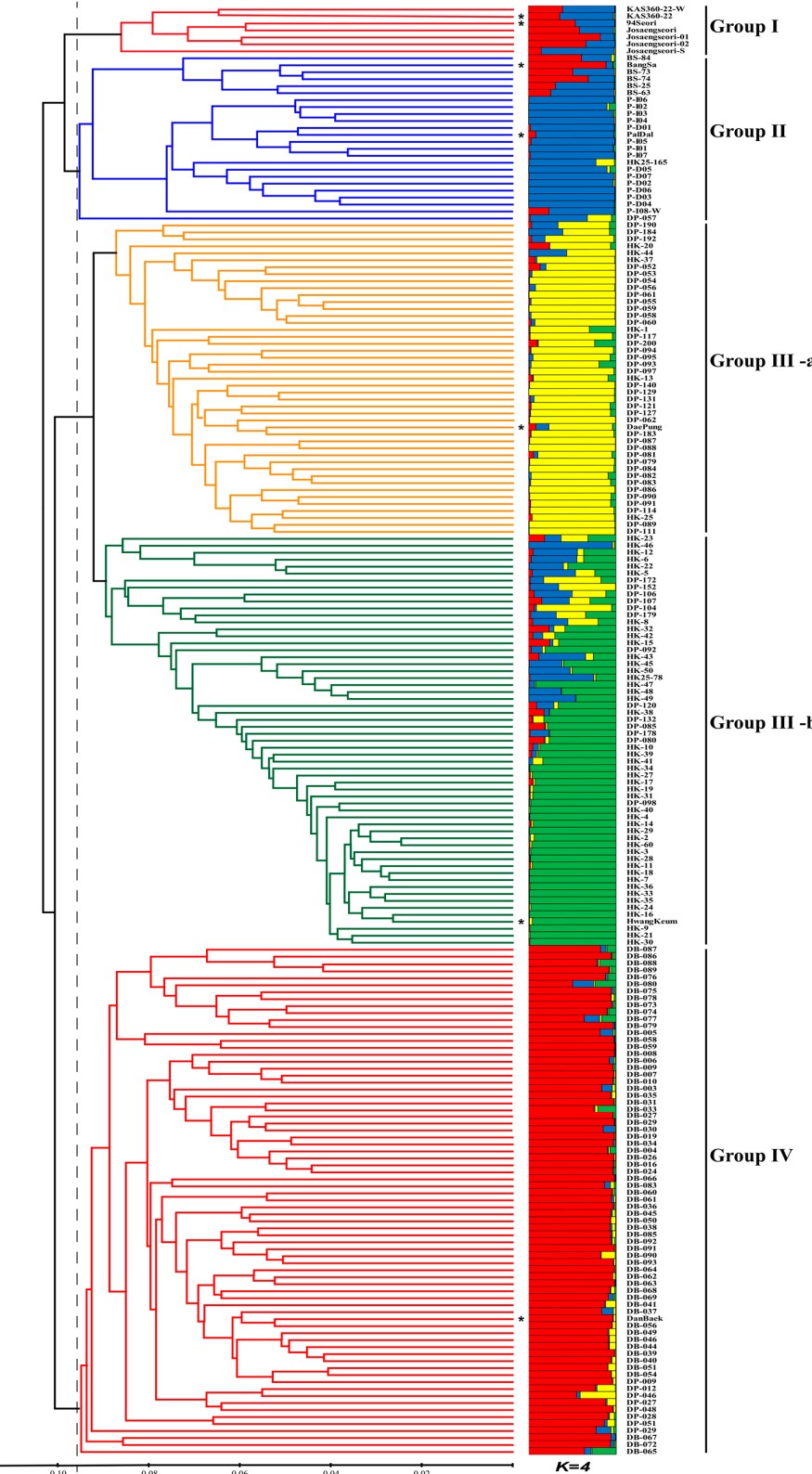

**Figure 4.** Dendrogram revealed by unweighted pair group method with arithmetic mean (UPGMA) cluster analysis and the population structure of 208 soybean MDP lines based on TRAP markers. * Indicated original cultivars.

### 3.4. AMOVA

An AMOVA of the 208 MDP lines based on TRAP markers was performed to analyze the distribution of inter- (among) and intra- (within) mutant population genetic diversity. According to the AMOVA, the estimated inter-mutant population variance was 11.320 (20.6%), while approximately 79.4% of the variation in all variance positions was attributed to intra-mutant population variance. These results indicate that the majority of the variance was intra-mutant populations. However, notwithstanding the estimation of variance in each intra-mutant population, the variation of intra-mutant population was mostly lower than inter-mutant population. The highest percentage of observed intra-mutant population genetic variation was that of DB populations (26.1%) and the lowest was intra KAS360-22 populations (0.4%). We also examined genetic differentiation between soybean MDP lines using Fst data estimated from pairwise comparisons. Fst values varied from 0.065 (KAS360-22 and 94seori) to 0.351 (HK and 94seori), with an average of 0.248 (Table 5).

**Table 5.** Analysis of molecular variance results and pairwise Fst values estimated from 208 soybean MDP lines.

| Source | Est. Var. | Percentage of variation (%) |
|---|---|---|
| Inter-mutant pops | 11.320 | 20.6 |
| Intra 94Seori pops | 0.820 | 1.5 |
| Intra KAS360-22 pops | 0.231 | 0.4 |
| Intra BangSa pops | 0.866 | 1.6 |
| Intra PalDal pops | 2.765 | 5.0 |
| Intra DanBaek pops | 14.324 | 26.1 |
| Intra DaePung pops | 14.252 | 26.0 |
| Intra HwangKeum pops | 10.342 | 18.8 |

| Control | 94Seori | KAS360-22 | BangSa | PalDal | DanBaek | DaePung | HwangKeum |
|---|---|---|---|---|---|---|---|
| 94Seori | – | | | | | | |
| KAS360-22 | 0.065 | – | | | | | |
| BangSa | 0.273 | 0.243 | – | | | | |
| PalDal | 0.312 | 0.270 | 0.285 | – | | | |
| DanBaek | 0.239 | 0.229 | 0.222 | 0.268 | – | | |
| DaePung | 0.276 | 0.245 | 0.258 | 0.226 | 0.129 | – | |
| HwangKeum | 0.351 | 0.343 | 0.322 | 0.270 | 0.254 | 0.126 | – |

Significance ($p < 0.001$) was determined based on 1000 iterations.

### 3.5. Association Analysis

Associations between 329 polymorphic fragments (16 combinations of TRAP markers) and 11 phenotypic traits of 208 soybean MDP lines were analyzed by three methods: (1) SFA; (2) a structure association analysis using a general linear model where population membership served as covariates (Q GLM); and (3) a composite approach in which the average relationship was estimated by kinship (*K*) and implemented under a mixed linear model (Q + K MLM). The average level of significance of each marker–trait association was based on the three different analyses are shown in Table 6. A total of 27 significant ($p < 0.001$) marker–trait associations (SMTAs) were detected using the three methods. The 157B + Sa4 primer combination was significantly associated with four phenotypic traits (GT, SA, PH, and NN), while 156A + Sa12, 156A + Sa4, 157B + Ga5, 157B + Ga3, 159A + Sa12, 159A + Ga5, and 159A + Ga3 were associated with one trait each: FC, PH, FC, PH, SHC, SI, and RN, respectively. The lowest calculated *P*-value using SFA was for the association of 157B + Sa12_6 with the FC trait ($p = 1.44 \times 10^{-12}$, $R^2 = 0.216$). Under the Q GLM model, the lowest *P*-value of a SMTA was that of B14 + Sa4_18 with PH ($P = 2.46 \times 10^{-8}$, $R^2 = 0.119$). Under the Q+K MLM model, the lowest SMTA *P*-value was observed for the association of B14+Sa4_18 with PH ($P = 9.81 \times 10^{-7}$, $R^2 = 0.107$) (Table 6). Among the three different analytical approaches (SFA, Q GLM, and Q + K MLM), the highest total number of SMTAs (626) was detected using SFA, followed by the Q GLM approach (178). The lowest number of SMTAs (143) was detected using the Q+K MLM approach; this number corresponded to 22.8% and 80.3% of the total number of SMTAs detected with SFA and Q GLM, respectively (Table S2). Six SMTAs

at $p < 0.0001$—two for GT (156A + Ga5_16 and 157B + Sa4_4), one for FC (157B + Sa12_6), one for SCC (157B + Sa12_20), one for PH (B14 + Sa4_18), and one for NN (B14 + Sa4_18)—were revealed by all three approaches when kinship and/or population structure was considered in this collection.

**Table 6.** Selection of 27 significant marker–trait associations (SMTA) markers based on three association analysis approaches.

| Trait | Marker | SFA [a] | $R^2$ | Q GLM [b] | $R^2$ | Q + K MLM [c] | $R^2$ | Average *p*-value |
|---|---|---|---|---|---|---|---|---|
| Maturity days | B14 + Ga5_28 | ** | 0.073 | ** | 0.044 | * | 0.044 | * |
| Growth type | 156A + Ga5_16 | ** | 0.116 | ** | 0.101 | ** | 0.111 | ** |
| | 157B + Sa4_4 | ** | 0.125 | ** | 0.102 | ** | 0.102 | ** |
| Flower color | 157B + Sa12_6 | ** | 0.216 | ** | 0.074 | ** | 0.072 | ** |
| | 156A + Sa12_17 | ** | 0.159 | * | 0.057 | * | 0.057 | * |
| | 157B + Ga5_9 | ** | 0.125 | * | 0.046 | * | 0.046 | * |
| Seed coat color | 157B + Sa12_20 | ** | 0.083 | ** | 0.083 | ** | 0.086 | ** |
| Seed hilum color | 159A + Sa12_28 | * | 0.055 | * | 0.058 | * | 0.063 | * |
| | 159A + Sa4_33 | ** | 0.111 | * | 0.058 | * | 0.057 | * |
| | 157B + Sa12_20 | * | 0.066 | * | 0.053 | * | 0.056 | * |
| Seed index | 156A + Ga3_7 | ** | 0.082 | ** | 0.054 | * | 0.054 | ** |
| | 159A + Sa4_32 | ** | 0.096 | * | 0.050 | * | 0.050 | * |
| | 156A + Ga3_1 | * | 0.065 | * | 0.046 | * | 0.046 | * |
| | 159A + Ga5_2 | ** | 0.139 | * | 0.044 | * | 0.044 | * |
| Stem anthocyanin | 157B + Sa4_7 | ** | 0.079 | ** | 0.047 | * | 0.044 | ** |
| Plant height | B14 + Sa4_18 | ** | 0.152 | ** | 0.119 | ** | 0.107 | ** |
| | B14 + Ga5_30 | ** | 0.173 | ** | 0.069 | * | 0.059 | ** |
| | 157B + Ga3_10 | ** | 0.135 | ** | 0.062 | * | 0.052 | * |
| | 157B + Sa4_4 | ** | 0.072 | * | 0.053 | * | 0.051 | * |
| | 156A + Sa4_6 | * | 0.056 | * | 0.048 | * | 0.049 | * |
| | 156A + Ga5_13 | ** | 0.100 | ** | 0.063 | * | 0.049 | * |
| Node number | B14 + Sa4_18 | ** | 0.121 | ** | 0.097 | ** | 0.087 | ** |
| | B14 + Ga5_30 | ** | 0.162 | ** | 0.078 | * | 0.066 | ** |
| | 157B + Sa4_4 | ** | 0.083 | * | 0.062 | * | 0.055 | * |
| | 156A + Ga5_16 | ** | 0.074 | * | 0.046 | * | 0.054 | * |
| Ramification number | 159A + Ga3_24 | * | 0.058 | * | 0.054 | * | 0.061 | * |
| | 159A + Sa4_32 | ** | 0.080 | ** | 0.072 | * | 0.058 | * |

[a] SFA: single factor analysis of variance. [b] Q GLM: general linear model using a Q population structure matrix. [c] Q + K MLM: mixed linear model using Q population structure and K kinship matrixes. * $p \leq 0.001$, ** $p \leq 0.0001$.

## 4. Discussion

In this study, we constructed an MDP from populations of 1695 gamma-irradiated mutants in two selection phases over $M_1$ to $M_{12}$ generations; first, in the $M_5$ generation, we selected 523 mutant lines exhibiting at least 30% superior agricultural characteristics, and, second, we eliminated redundant morphological phenotypes in the $M_{12}$ generation (Figure 1). Finally, we constructed 208 MDP lines and investigated 11 agronomic traits. Our collection strategy for selecting MDP lines differed in some respects from the general core-collection method. With the latter approach, a core collection assembled from an existing collection is chosen to represent the genetic and phenotypic diversity of the larger collection without overlapping phenotypes [30]. Such an approach has become accepted as an efficient tool for improving the conservation of many crops [31,32]. In our study, we similarly eliminated overlapping phenotypes from our collected MDP lines in the second selection phase, but we considered specific changed agronomic characteristics of individual mutants rather than their representation of the original populations.

Our examination of agronomic traits in the MDP lines revealed a variety of DF, MD, GT, FC, SA, PH, NN, and RN phenotypes as well as those related to seed traits, such as SCC, SHC, and SI (Table 2, Figure 2, Table S1). We also observed changes in phenotypes between MDP lines and their wild types (Figure 3, Table S3, Figure S1). The FAO/IAEA mutant variety database (MVD, http://mvd.iaea.org)

includes 174 publicly released soybean mutants. These mutants have various desirable agronomical and biochemical characteristics, such as an improved maturity date, yield, protein content, fatty acid content, and changed seed/stem color, with approximately 62% of released mutants mainly selected for their altered maturity dates and yields. In our phenotypic evaluation of the 208 MDP lines, we detected a wider variety of changes to the quantitative traits, including SI, PH, NN, and RN (Figure 3), as well as to the qualitative traits, such as FC, SCC, and SHC (Figure S1). According to our previous study, in addition, some of DB- and DP-derived mutants in the MDP lines had changed compositions of fatty acids, including linolenic acid and oleic acid [33]. Given all of these results, our MDP lines may be useful resources as a genetic diversity pool for soybean breeding.

To investigate genetic relationships among the 208 MDP lines, we evaluated DNA polymorphism patterns in these lines using TRAP markers. In the rapid, efficient PCR-based TRAP marker system, expressed sequence tag database information and bioinformatics tools are used to generate polymorphic markers around targeted candidate gene sequences. Previous studies of lettuce (*Lactuca sativa*) [34], sugarcane (*Saccharum officinarum*) [35], spinach (*Spinacia oleracea*) [29], geranium (*Pelargonium inquinans*) [36], sunflower (*Helianthus annuus*) [37], and faba beans (*Vicia faba*) [16] have demonstrated that TRAP markers are useful for assessing genetic diversity. Using this system in the present study, we PCR-amplified 551 fragments with 16 primer combinations and observed considerable variation in the percentage of polymorphic amplicons among primer pairs—from 32.35% to 84.00% (Table 4). In a study of faba beans, Kwon et al. [16] obtained 221 amplified fragments with 12 TRAP primer combinations and observed an average polymorphism rate of 55.2%. In the present study, we observed a polymorphism level of 59.7% among 551 amplified fragments. In contrast, a study of sugarcane detected a polymorphism rate of 74% from 925 amplified fragments [17], a level much higher than in the soybeans (*Glycine max*) and the faba beans. Compared with the results of previous studies of soybeans based on ISSR and RAPD techniques [38,39], the use of the TRAP system yielded more DNA fragments per primer combination. A previous AFLP analysis generated an average of 40 to 50 DNA fragments per primer pair [40,41], similar to the outcome of our TRAP analysis. The present results demonstrate that the TRAP marker system is a simple yet powerful technique for estimating soybean genetic diversity.

To reveal relationships among the 208 MDP lines, we constructed a UPGMA-based dendrogram using the TRAP marker data. On the basis of genetic distances, the 208 MDP lines clustered into four groups. An analysis of the population structure based on an ad-hoc statistic ($\Delta K$) likewise divided the MDP lines into four groups. These results indicate that four genotype-based sub-populations are present in the 208 MDP lines (Figure 4) that largely correspond to their wild-type cultivars. As denoted by different colors, the main membership composition of the four groups and their subgroups is as follows: Group I including two wild types (94seori, and KAS360-22) possessed 52% red and 46% blue; Group II including BS and P was 80% blue; Group III-a including DP was 88% yellow; Group III-b including HK was 71% green; and Group IV including DB was 90% red. In a previous genetic diversity analysis based on 20 SSR markers, 91 Korean soybean cultivars were divided into seven groups at a genetic distance of 0.81. In that study, HK and P were clearly separated, but three cultivars (BS, DB, and DP) grouped together [42]. Using TRAP markers in the present study, we were able to better resolve groups of wild-type cultivars. In addition, we performed an AMOVA to separate the total molecular variance of the mutants into inter- and intra-population components (Table 5) and assessed their significance using permutational testing procedures. Overall, based on the dendrogram and population structure, 201 mutant lines grouped with their wild type except 29 (14%) mutant lines, including 22 DP- and 7 HK-mutants. Nevertheless, these mutant lines also possessed their genetic membership according to population structure. In AMOVA, all intra-mutant population also showed lower variation than inter-mutant population except for two populations, DB- and DP-, since DB- and DP- had most large mutant lines, 64 and 60, respectively. A similar result was described by Lee et al. [20]. Each of the ten wild types was clustered with their $M_1$ generation mutants by gamma radiation in faba bean. However, the genetic variation of the mutants is not much higher than among cultivars

or accessions. Although TRAP markers have most commonly been used for genetic mapping and dendrogram studies, they have also recently been applied to detect DNA mutations. Because of their many advantages, including simplicity, reliability, moderate throughput, and ease of sequencing of selected bands, TRAP markers have been used widely in plants. For example, the TRAP system has been used to study genetic variability induced by gamma ray treatments in sugarcane [43] and sorghum (*Sorghum bicolor*) [19]. Lee et al. [20] recently exploited a TRAP marker to estimate the frequency of mutations induced by gamma rays in an $M_1$ generation of faba bean. The 242 amplified fragments obtained using eight primer combinations had an average polymorphism rate of 66.7%, which is higher than the percentage in our study because they used early generation. TRAP markers have several advantages over other types of markers: they are easy to use (like RAPDs), high in polymorphisms (like AFLPs), and their primers can be readily designed from known sequences of putative genes [44].

In association mapping, false discoveries are a major concern and can be partially attributed to spurious associations caused by population structure and unequal relatedness among individuals. Two major approaches, namely, GLM and MLM, are used to study marker–trait associations. The number of SMTAs detected by GLM is generally much higher than that revealed by MLM [45]. GLM-based studies of marker–trait associations consider only the Q matrix generated during the study of population structure. In contrast, MLM simultaneously accounts for both population structure and kinship (genetic relatedness among individuals) and is hence more reliable. In the present study, the GLM method (Q) uncovered 178 SMTAs between the 11 phenotypic traits and 27 TRAP markers. Using the MLM method (Q + K), 143 SMTAs involving 27 TRAP markers were identified (Table 6, Table S2). These results confirm a previous observation that the number of SMTAs estimated with GLM is higher than that uncovered with MLM [46,47]. Most interestingly, the three approaches considering kinship and/or population structure in the MDP collection in this study revealed six SMTAs at $p < 0.0001$ in all approach methods. These six SMTAs involved five agronomic traits: GT (2), FC (1), SCC (1), PH (1), and NN (1).

## 5. Conclusions

In this study, we successfully constructed soybean MDP lines and compared their agronomic traits. We also performed the first-ever study of genetic diversity and relationships using the TRAP marker system in soybean. To examine MDP genetic diversity and relationships, we performed dendrogram, population structure, and molecular variance analyses based on their TRAP genotypes. Finally, we uncovered six SMTAs ($p < 0.0001$) involved with TRAP genotypes and agronomic traits using three association mapping methods (SFA, Q GLM, and Q + K MLM). Our results can serve as a foundation for future research on genotype–phenotype interactions in large mutant populations.

**Supplementary Materials:** The following are available online at http://www.mdpi.com/2073-4395/10/2/253/s1, Figure S1: Changes in five qualitative-trait phenotypes of 208 MDP lines; Figure S2: ΔK values for different numbers of populations (*K*) assumed in the STRUCTURE analysis, Table S1: Agronomic characteristics of the 208 soybean MDP lines used in this study; Table S2: Comparison of the number of SMTAs ($p < 0.01$) based on three analytical approaches (SFA, Q GLM, and Q + K MLM); Table S3: Comparison of phenotypes and genotypes among seven original cultivars.

**Author Contributions:** Conceptualization, D.-G.K., J.I.L., and S.-J.K.; methodology, D.-G.K., J.I.L., and S.-J.K.; software, D.-G.K. M.-K.L.; validation, D.-G.K., J.-B.K., C.-H.B., and S.-J.K.; investigation, D.-G.K., J.M.K.; resources, D.-G.K., M.J.H.; data curation, D.-G.K., M.-K.L., N.N.H.; writing—original draft preparation, D.-G.K.; writing—review and editing, J.I.L., S.-J.K.; visualization, M.J.H., J.-B.K.; supervision, C.-H.B., S.-J.K.; funding acquisition, J.-B.K., S.-J.K. All authors have read and agreed to the published version of the manuscript.

**Funding:** This research was funded by the Radiation Technology R&D Program (NRF-2017M2A2A6A05018538) through the National Research Foundation of Korea funded by the Ministry of Science and ICT.

**Acknowledgments:** We thank Edanz Group (www.edanzediting.com/ac) for editing the English text of a draft of this manuscript.

**Conflicts of Interest:** The authors declare no conflict of interest.

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
