# Peer review of "Construction of Soybean Mutant Diversity Pool (MDP) Lines and an Analysis of Their Genetic Relationships and Associations Using TRAP Markers"

_agronomy, doi:10.3390/agronomy10020253_

Round 1
Reviewer 1 Report
With respect to the earlier version I do not note many changes. (By the way the WEB address of the FAO/IAEA Mutant Varieties Database has not been corrected. It is indeed https://mvd.iaea.org/)
Of course there has been a general slight improvement of the paper, but still the main question has not been addressed: genetic analysis
The authors still refer to methods of phylogenetic analysis, which is not their case. They deal with an artificial population in which an undetermined amount of diversity has been artificially eliminated due to the selection method utilized. The term "phylogenetic" appearing here and there in the document is absolutely inappropriate. Moreover the markers they used are not neutral, since based on expressed genomic areas.
A genetic analysis based on phylogenetic or population tools in in this case a mistake. To describe the genetic diversity in this case a PCoA would have been much more informative and appropriate
Author Response
Reviewer #1
With respect to the earlier version I do not note many changes. (By the way the WEB address of the FAO/IAEA Mutant Varieties Database has not been corrected. It is indeed https://mvd.iaea.org/)
[Response] As suggested, we have changed web address of the FAO/IAEA MVD site.
Of course there has been a general slight improvement of the paper, but still the main question has not been addressed: genetic analysis
The authors still refer to methods of phylogenetic analysis, which is not their case. They deal with an artificial population in which an undetermined amount of diversity has been artificially eliminated due to the selection method utilized. The term "phylogenetic" appearing here and there in the document is absolutely inappropriate. Moreover the markers they used are not neutral, since based on expressed genomic areas.
[Response] A purpose of this research was measuring mutation frequencies comparing with different wild types and mutant lines. Of course, our MDP lines were constructed artificial two selection process for collection of different phenotypes compared with their original cultivars, but these process are essential for finding to novel resources in mutation breeding.
As reviewer’s mentioned, phylogenetic analysis is not suitable for our study. Although we slightly agree that the term “phylogenetic” is not suitable, but clustering or dendrogram methods according to molecular marker based on expressed genomic areas has been used in many previous mutation breeding research (Khan et al., 2010; Malek et al., 2014; Sharma et al., 2016; Mirajkar et al., 2017; Lee et al., 2019). In addition, molecular marker based on expressed genomic region such as EST-SSR, TRAP, SNPs, which has been studied for association of the agronomic traits(Simko and Hu, 2008; Yue et al., 2009; Kwon et al., 2013; Ukoskit et al., 2019). Thus, we have maintained our genetic analysis approach, but changed term “phylogenetic tree” to “Dendrogram” in maintext and Figure 4.
As our TRAP marker results, although MDP has been two selection process before TRAP screening, genetic diversity and polymorphic levels were showed similar to natural accessions(germplasm) levels (Kwon et al., 2010). Also, dendrogram (phylogenetic tree) and population structure, we have defined various genetic changes by gamma irradiated mutant lines. These results indicate that MDP could be fully available as a genetic resources pool. However, the genetic changes of mutants from each wild type was not much greater than genetic differentiation of among wild types. Therefore, before deeper research such as whole genome re-sequencing or GBS, the genetic diversity research was valuable for our further research to select some mutant lines. As you know, sequencing cost still expensive for whole population.
Khan et al., (2010) Genetic variability in mutated population of sugarcane clone NIA-98 through molecular markers (RAPD and TRAP). Pak. J. Bot., 42(1): 605-614.
Malek et al., (2014) Morphological characterization and assessment of genetic variability, character association, and divergence in Soybean mutants. The Scientific world Journal, http://dx.doi.org/10.1155/2014/968796
Sharma et al., (2016) Polymorphism analysis in advanced mutant population of oat (Avena sativa L.) using ISSR markers. Physiol. Mol. Biol. Plants, 22(1): 115-120.
Mirajkar et al., (2017) TRAP and SRAP molecular marker based profiling of radiation induced mutants of sugarcane (Saccharum officinarum L). Plant Gene, 9: 64-70.
Lee et al., (2019) Utility of TRAP markers to determine indel mutation frequencies induced by gamma-ray irradiation of faba bean (Vicia faba L.) seeds. International Journal of Radiation Biology 1-12.
Kwon et al., (2010) Genetic diversity and relationship among faba bean (Vicia faba L.) germplasm entries as revealed by TRAP markers. Plant Genetic Resources, 8: 204-213.
Simko and Hu, (2008) Population structure in cultivated lettuce and its impact on association mapping. J. Amer. Soc. Hort. Sci. 133(1): 61-68.
Yue et al., (2009) Genetic diversity and relationships among 177 public sunflower inbred lines assessed by TRAP markers. Crop Science, 49: 1242-1249.
Kwon et al., (2013) Genome-wide association of 10 horticultural traits with expressed sequence tag-derived SNP markers in a collection of lettuce lines. The Crop Journal (2013) 25-33.
Ukoskit et al., (2019) Detection and validation of EST-SSR markers associated with sugar-related traits in sugarcane using linkage and association mapping. Genomics, 111(1): 1-9.
A genetic analysis based on phylogenetic or population tools in in this case a mistake. To describe the genetic diversity in this case a PCoA would have been much more informative and appropriate
[Response] yes, we are agree PCoA is appropriate analysis for our research and we performed this before. But, the result of PCoA showed similar with clustering analysis as below.
Since this results are duplicated of the dendrogram, so we have omitted PCoA result in present study.

Reviewer 2 Report
The title and aims of the research are coherent to the scope of the journal. The abstract is clearly described and comprehensive. The introduction is properly composed. Material, Methods, and protocols are standard. Research is technically well organized and provides novel original data.
Overall, the results are presented in a comprehensible manner, with supporting charts and graphs to illustrate the statistical analyses. In a discussion, the obtained results should be better compared with analogical researches.
The authors used only a few references from the period of the past 5 years. I recommend actualizing the list of references.
I recommend also to characterize phenotype – morphological and agronomic traits.
Author Response
Reviewer #2
The title and aims of the research are coherent to the scope of the journal. The abstract is clearly described and comprehensive. The introduction is properly composed. Material, Methods, and protocols are standard. Research is technically well organized and provides novel original data.
Overall, the results are presented in a comprehensible manner, with supporting charts and graphs to illustrate the statistical analyses. In a discussion, the obtained results should be better compared with analogical researches.
[Response] We thank the reviewer for the positive feedback.
The authors used only a few references from the period of the past 5 years. I recommend actualizing the list of references.
[Response] As suggested, we have changed list of references excluded less relevant to this study.
I recommend also to characterize phenotype – morphological and agronomic traits.
[Response] As suggested, we classify the 11 traits with referring to the previously study (Mudibu et al., 2012) and changed description in Materials and Methods section 2.1 as follows: “ Overall, we selected 208 genetically fixed mutant lines (201 mutants with their wild types), which we designated as the mutant diversity pool (MDP) and were assessed the 4 agronomic traits: days of flowering (DF), maturation date (MD), 100-seed weight (100SW), node number (NN), and 7 morphological traits: growth type (GT), flower color (FC), seed coat color (SCC), seed hilum color (SHC), stem anthocyanin (SA), plant height (PH), and ramification number (RN).” (revised manuscript; Line 101-105)
Justin Mudibu et al., (2012) Effect of gamma irradiation on morpho-agronomic characteristics of Soybeans (Glycine max L.). American Journal of Plant Sciences, 3: 331-337.

Reviewer 3 Report
This paper reports on the construction of MDP lines and their genetic relationships. The study represents a great volume of work, is well-written and provides good insight to genetic relationships of several traits.
Following are questions/concerns:
Line 89. Can you provide citations or other documentation for these cultivars? Lines 90-101. I had some difficulty understanding the origin of these lines/selections until I read lines 161-170. I suggest moving lines 161-170 to the section 2.1. Line 165-170. How were these characteristics determined? Line 168. How were “redundant phenotypes” identified? Many quantitative traits are evenly distributed over a range of phenotypes. Line 174. You use acronyms in text without defining them. Line 177-178. The morphological (qualitative) traits do not follow a Gaussian (normal) distribution. Line 184. HK-derived mutants are not identified in Fig 3. Line 195. Table 2. How, where and when were these quantitative traits measured? “100 seed weight” is equal to “seed index”, which is better terminology for this trait. Line 308 and 349. Add scientific names. Line 330. “varieties” should be “cultivars” Line 342. Consider starting new paragraph with “The mutation breeding..” so that paragraph will not be so long. Also, seems to be break in discussion
Author Response
Reviewer #3
This paper reports on the construction of MDP lines and their genetic relationships. The study represents a great volume of work, is well-written and provides good insight to genetic relationships of several traits.
Following are questions/concerns:
Line 89. Can you provide citations or other documentation for these cultivars?
[Response] As requested by the reviewer, we provided citation containing genetic information of 5 original cultivars excluded KAS360-22 (landrace) and 94seori in revised manuscript; Line 88.
Lines 90-101. I had some difficulty understanding the origin of these lines/selections until I read lines 161-170. I suggest moving lines 161-170 to the section 2.1.
[Response] We thank the reviewer for this valuable suggestion. As suggested, we have moved the lines 161-170 to the section 2.1 as follows: “A total of 1,000 seeds each of one soybean landraces, KAS360-22, and six representative Korean soybean cultivars [27], 94Seori, BangSa (BS), PalDal (P), DanBaek (DB), DaePung (DP), and HwangKeum (HK), were irradiated with 250 Gy of gamma rays using a 60Co gamma-irradiator (150 TBq capacity; ACEL, Ottawa, ON, Canada) at the Korea Atomic Energy Research Institute in 2008. The irradiated M1 and control (non-irradiated) seeds were immediately sown in the research field of the Advanced Radiation Technology Institute. To construct MDP lines, we sowed 1,000 irradiated seeds of each of seven soybean cultivars and harvested a total of 1,695 M1:2 individual seeds, only excluding those exhibiting growth aberrations, such as stunted growth, pollen sterility, and no germination due to the degree of radiosensitivity (Figure 1). Next, we generated 1,695 individual gamma-irradiated mutants during M1–M5 generations by single-seed descent and then continued as bulks to M12 generation. In a first selection phase, we selected 523 mutant lines from the M5 generation that possessed at least 30% superior agricultural characteristics related to various environmental factors, such as grain yield, growth type, and climate adaptability. In a second selection phase, we investigated the morphological phenotypes of the 523 mutant lines in the M12 generation to eliminate redundant phenotypes. Overall, we selected 208 genetically fixed mutant lines (201 mutants with their wild types), which we designated as the mutant diversity pool (MDP) and were assessed the 4 agronomic traits: days of flowering (DF), maturation date (MD), seed index (SI), node number (NN), and 7 morphological traits: growth type (GT), flower color (FC), seed coat color (SCC), seed hilum color (SHC), stem anthocyanin (SA), plant height (PH), and ramification number (RN).” (Line 87-105)
Line 165-170. How were these characteristics determined?
[Response] In this context, we have applied to various changed agronomical and morphological traits values compared with their each original cultivars. As above described in lines 165-170 (it moved to section 2.1), the phenotype analysis and selection criteria were considered various changed traits such as 100 seed weight (38% ~ 213% vs cultivar), days of flowering (84% ~ 133%), maturity days (85% ~ 122%), plant height (62% ~ 370%), node number (73% ~ 286%), ramification number (65% ~ 194%) and extremely changed morphological traits. Therefore, all mutant lines has changed its phenotype more than one compared with original cultivars. For example, BS-derived 5 mutant lines were exhibited mainly changed 2 trait values such as maturity days, stem anthocyanin traits and P-derived 14 mutants changed growth type, seed coat color, stem anthocyanin traits. More detailed information presented in Supplementary Table S1.
Line 168. How were “redundant phenotypes” identified? Many quantitative traits are evenly distributed over a range of phenotypes.
[Response] The redundant phenotype were eliminated in a second selection phase (M12 generation), which means are selected 208 MDP lines without showing redundant phenotypes (almost same morphological characteristics compared with their each original cultivar) from 523 first-selected mutant lines. And above mentioned, all mutant lines has changed its phenotype more than one compared with original cultivars.
Line 174. You use acronyms in text without defining them.
[Response] The DF and MD are mentioned in the section 2.1(materials and methods), mutant line numbers such as S87, S88, have added information at first mentioned part as follows: “DF ranged from 42 (mutant numbers; S87, S88, and S149)~” (Line 170)
Line 177-178. The morphological (qualitative) traits do not follow a Gaussian (normal) distribution.
[Response] As noted by the reviewer, we have changed “11 agronomic and morphological traits” to “6 quantitative traits” (Line 173)
Line 184. HK-derived mutants are not identified in Fig 3.
[Response] HK(HwangKeum)-derived mutants data were showed at the rightmost of each box plots.
Line 195. Table 2. How, where and when were these quantitative traits measured? “100 seed weight” is equal to “seed index”, which is better terminology for this trait.
[Response] As requested by the reviewer, we added the basic information at Table 2 footnote. And also we have changed “100 seed weight” to “seed index” in Table 2 and maintext.
Line 308 and 349. Add scientific names.
[Response] As suggested, we have added scientific names at each plants. (Line 304-305, 346)
Line 330. “varieties” should be “cultivars”
[Response] we have changed “91 Korean soybean varieties” to “91 Korean soybean cultivars” (Line 328)
Line 342. Consider starting new paragraph with “The mutation breeding.” so that paragraph will not be so long. Also, seems to be break in discussion
[Response] We thank the reviewer for the comment. We agree with the reviewer’s point and deleted this paragraph.

Round 2
Reviewer 1 Report
The paper has improved since the previous version and one major fault (phylogenetic analysis) has been fixed.
Still I hve doubts about the general interest in the experiments and their results
This manuscript is a resubmission of an earlier submission. The following is a list of the peer review reports and author responses from that submission.
Round 1
Reviewer 1 Report
The introduction should briefly show what is already known and then focus more on what is not known and why it is worth studying. Although the authors start to explain the importance of the mutation breeding, the link to this study is only one of the alternatives. How does the current work help to improve tolerance to the harmful environment?
Physiological interpretations are weak. I doubt the applicability of the results.
The content of the manuscript is interesting and important for a wide range of readership in this research field. The article is well organized. The abstract of the paper is well compressed, realistic, understandable.
Physiological interpretations are weak. I doubt the applicability of the results.
A close collaboration between agronomists, plant physiologists, geneticists, biotechnologists is the pressing need and must be envisioned in order to address crop performance under the prevailing extreme climatic conditions.
Arguments need clearer and tighter presentation. The understanding of mechanisms is very limited, as it is restricted to papers that have a particular view and deliberately ignore alternatives, and does not present a balanced view of the evidence.
My major critical remarks refer to the complexity of the paper. Paper is focused mostly on molecular aspects, not on applied, agronomy research.
Authors could better discuss applied aspects of their research and improve conclusions and perspectives of their research.
Reviewer 2 Report
This paper reports on the assembly of a collection of morphological mutants of soybean obtained by gamma irradiation. The aim is supposedly to have increased variation to be used in breeding programs. Morphological characters selected are examined statistically and their variation is reported. Also genetic variation is studied by a combination of TRAP markers. In my opinion there is a clear fracture between the morphological analysis and the genetic one.
The authors apply some genetic analysis methods to the data obtained by the codominant markers used. Here I identify a problem. The mutant pool was selected on a morphological basis, that is favoring only those morphotypes that were considered "supeior", and founded on quantitaive characters. This has probably eliminated most of the "hidden" variation, that is the variation not associated to the morphological one. The analysis methods used, instead, are all based on the assumption that the analyzed population is genetically cohesive and not under strong selection effect. Moreover, the authors state they have eliminated "morphologically redundant" plants, this further reduced the genetic variation and increased selection pressure.
In my opinion the whole genetic analysis adds no significant information; the morphological analysis alone is sufficient to describe the 208 MDP genotypes collection. A paper dealing only with the morphological description and analysis would be much more focused and of interest.
Some detailed notes:
Line 51 the address for the FAO/IAEA Mutant Variety Database is incorrect. Please correct
Lines 60-73 Describing the TRAP markers many sentences appear to have been quite completely copied by other papers. The authorship should be better cited and the concepts reformulated
Lines 177-179 The authors state that the characters scored are distibuted by the Gaussian distribution, but a sight to fiigure 2 makes me doubt that the statement is correct (see DF and MD); moreover some characters cannot be normally distributed since qualitative (GT, FC, SCC, SHC)
Line 252 "Associations between 329 genotypes"... I do not understand. I thaught that the authors dealt with 208 SSD gentypes derived from a much wider number of nutants. Did they intend haplotypes? and if so, how did they determine them?
Lines 291-303. The authors often refer to the FAO/IAEA mutant variety database and consider their collection similar. In fact, this is not a collection of mutants, because only quantitaive traits were considered during the selection. Since most quantitative traits are the result of the action of several genes modulated through the metabolic networks, one cannot say that a taller plant or a higher !00 seed weight are the result of a mutation. A mutation may be supposed somewhere with no certainty